# Deep Kernel learning for reaction outcome prediction and optimization
Sukriti Singh ✉ & José Miguel Hernández-Lobato ✉

Recent years have seen a rapid growth in the application of various machine learning methods for reaction outcome prediction. Deep learning models have gained popularity due to their ability to learn representations directly from the molecular structure. Gaussian processes (GPs), on the other hand, provide reliable uncertainty estimates but are unable to learn representations from the data. We combine the feature learning ability of neural networks (NNs) with uncertainty quantification of GPs in a deep kernel learning (DKL) framework to predict the reaction outcome. The DKL model is observed to obtain very good predictive performance across different input representations. It significantly outperforms standard GPs and provides comparable performance to graph neural networks, but with uncertainty estimation. Additionally, the uncertainty estimates on predictions provided by the DKL model facilitated its incorporation as a surrogate model for Bayesian optimization (BO). The proposed method, therefore, has a great potential towards accelerating reaction discovery by integrating accurate predictive models that provide reliable uncertainty estimates with BO.

Chemical reaction optimization is central to organic synthesis and has largely been based on chemical intuition[1]. During optimization, the aim is to maximize the reaction outcome (e.g., yield and/or enantiomeric excess) by identifying suitable experimental conditions[2]. This involves evaluating a multidimensional chemical space comprising of reaction variables such as catalyst, solvent, substrate, additive, time, temperature, concentration, etc.[3,4]. Owing to the complexity of this problem, several data-driven approaches have been employed for efficient exploration of the chemical space[5–7].

The estimation of reaction outcome is of great importance in reaction development. It could enable chemists to identify, for instance, low-yield reactions prior to wet-lab experiments, thereby saving time and resources. Machine learning (ML) has shown an impressive degree of success in many areas of chemistry[8–11]. Earlier efforts toward reaction outcome prediction use hand-crafted features such as physical organic descriptors and molecular fingerprints[12,13]. Conventional ML methods, particularly random forests, perform extremely well with these non-learned representations. Recently, the advances in deep learning (DL) have led to the development of new molecular representations[14]. These are learned directly from molecular structures like simplified molecular input line entry specifications (SMILES) and molecular graphs. The chemical language models (LMs) and graph neural networks (GNNs) trained using these string or graph-based representations have displayed great potential in reaction outcome prediction[15–18].

The reaction outcome prediction augmented with uncertainty quantification is expected to find superior utility during reaction optimization[19]. As an example, Bayesian optimization (BO) works with the uncertainty estimates to suggest new experiments in a search for optimal reaction conditions[20]. While the quantification of uncertainty using the above-mentioned ML methods might not be straightforward, the uncertainty-awareness of Gaussian processes (GPs) is well-known[21,22]. GPs are generally preferable to quantify uncertainty in low-data situations and are frequently used as surrogate models in BO[23,24]. The kernel for the GP model is usually fixed, implying that GPs are unable to learn representations from the data. Therefore, training GPs on commonly used molecular inputs such as SMILES and graphs is challenging. On the other hand, neural networks (NNs) can directly learn feature representations from the aforesaid inputs and convert them into a continuous vector. This representation learning strength of NNs can be combined with the uncertainty quantification of GPs using deep kernel learning (DKL)[25]. Thus, DKL can provide an advantage over standard GPs due to the ability of NNs to learn better and more flexible representations[26].

Previous ML models for reaction outcome prediction are suitable for either the nonlearned (e.g., molecular descriptors, fingerprints) or learned (e.g., SMILES and graphs) representations of molecular encodings[27–29]. In this work, we present an ML model that works well with both kinds of molecular inputs. It integrates NNs with GPs within a DKL framework to obtain accurate reaction outcome predictions with corresponding prediction uncertainties. This is one of the earliest examples of the application of DKL for a chemistry task[30]. To further demonstrate the usefulness of our method, we combine the uncertainty estimates provided by deep kernel learning with BO to optimize the reaction outcome. This would enable on-the-fly feature learning using unstructured and/or high-dimensional

Department of Engineering, University of Cambridge, Cambridge, UK. ✉e-mail: sukriti243@gmail.com; jmh233@cam.ac.uk

**Fig. 1 | A general scheme of Buchwald–Hartwig cross-coupling reaction.** The reaction is comprised of all possible combinations of 15 aryl halides, 4 catalysts, 3 bases, and 23 additives, resulting in a total of 3955 reactions. The reaction yield ranges from 0 to >99.9.

molecular data during BO[31]. Thus, we believe that the proposed method would broaden the application of GPs in reaction development and reactivity prediction.

## Results and discussion

To demonstrate our approach, we selected a catalytic transformation with high practical utility, namely, the Buchwald–Hartwig cross-coupling (Fig. 1)[32]. The reaction space, as available from high-throughput experimentation, involves all possible combinations of 15 aryl halides, 4 ligands, 3 bases, and 23 additives. The dataset therefore, consists a total of 3,955 reactions with their corresponding experimental yields. Previous studies have reported a yield prediction model on this dataset using various learned and nonlearned molecular representations[33]. The nonlearned representations are manually constructed by a human expert using the domain knowledge (physical organic descriptors, molecular fingerprints, etc.). The deep learning models (e.g., transformers and GNNs), on the other hand, directly learn representations from molecular structures such as SMILES and graphs[34–37]. The choice of ML model is primarily decided by the input representation and vice versa[38]. In the following sections, we briefly describe our approach and show how it can be used with both the learned and nonlearned representations.

### Deep Kernel learning with nonlearned molecular representations

In the early stages of reaction development, only a small amount of data is usually available. The nonlearned representations crafted using domain knowledge generally work well in low-data regimes. However, they typically fail to provide better performance than the learned representations when sufficient data is available[17,18]. On account of this, we decided to provide additional flexibility to the nonlearned representations by incorporating them into a DKL framework. Here, a fully connected NN is used to extract features from the nonlearned input representation, followed by a GP for yield prediction. An overview of the approach is shown in Fig. 2. The commonly used nonlearned representations include molecular fingerprints, which are sparse and high-dimensional bit vectors, and physical organic descriptors, which are relatively low-dimensional. We consider the following three molecular representations independently as an input to the DKL model: (1) molecular descriptors, (2) Morgan fingerprints[39], and (3) differential reaction fingerprint (DRFP)[40] (Supplementary Section 1). The molecular descriptors obtained from density functional theory (DFT) computations include features describing the electronic and spatial properties of molecules. To obtain the representation of the reaction, the features of all reactants are concatenated together, resulting in a total of 120 molecular descriptors (Supplementary Section 1.1) (Fig. 2). Next, Morgan fingerprints for each reactant are computed as a 512-bit vector with radius 2. The reaction representation is a concatenation of fingerprints for individual reaction components, giving a 2048-dimensional bit vector (Supplementary Section 1.2). Lastly, we use DRFP (with 2048 bits), which is constructed using reaction SMILES as an input and returns a binary fingerprint for the reaction (Supplementary Section 1.3).

As described in the previous section, the DKL model consists of a NN for feature learning and a GP for making predictions. In the case of nonlearned molecular inputs, we use a feed-forward NN with two fully connected layers as a feature extractor (Fig. 2). The reaction representation is first passed through the NN to obtain an embedding vector. The embedding is then passed as an input to the GP, which provides the prediction with uncertainty estimates. The resulting DKL model is trained by jointly optimizing all the NN parameters and GP hyperparameters using the log marginal likelihood of the GP (Eq. 4) as the objective function.

### Deep Kernel learning with learned molecular representations

Molecular graphs have been widely used as an input representation to GNNs for various chemistry-related tasks[41,42]. The GNNs can automatically capture relevant features from the molecular graph, which can then be utilized for predictions. In view of this, we build our DKL model using GNN as a feature extractor (Supplementary Section 2), and a GP for the task of yield prediction. An overview of the method is provided in Fig. 3b. For this purpose, the molecules are represented as an undirected molecular graph $G = (V, E)$, with atoms as nodes $V$ and bonds as edges $E$. The nodes and edges have their own set of features (Fig. 3a). The node features $x_v$ correspond to the atom features of heavy atoms (e.g., C, N, Cl, Br, and O). Examples of atom features are the atom type, hybridization, chirality, formal charge, etc. Similarly, the edge features $e_{vw}$ are associated with bond features for the bond between the $v^{th}$ and $w^{th}$ atoms. Bond features include bond type, whether the bond is conjugated or part of a ring, stereochemistry, etc. The open-source package RDKit[43] is used to compute all the atom and bond features.

For each reactant, once the graph with atom and bond features is built, we use this as an input to the message-passing neural network[44] to learn the graph embedding (Fig. 3b). For this purpose, we perform multiple message-passing steps to get the node representation, where an edge network serves as a message function and a gated recurrent unit (GRU) as an update function. For the readout step, a set2set model[45] is applied for global pooling over the node representation vectors. This provides a graph representation vector $\mathbf{r}$, invariant to the node order. Finally, the graph representation vector of individual reaction components $\mathbf{r}_{arylhal}$, $\mathbf{r}_{ligand}$, $\mathbf{r}_{base}$, $\mathbf{r}_{additive}$ are summed to obtain the composite representation for the reaction (Fig. 3b). Thus, the reaction representation becomes invariant to the order of reactants.

The reaction embedding as obtained from the GNN is first processed by a feed-forward neural network (FFNN), the output of which is subsequently used as an input to the base kernel of GP (Fig. 3b). Thus, the resulting DKL model has two components: the GNN for feature learning and the GP for predictions. The model is trained in an end-to-end manner, where all the parameters of the deep kernel are learned jointly by maximizing the log marginal likelihood $\mathcal{L}$ of the GP (Eq. 4). For making predictions at test time, we compute the posterior predictive distribution (Eq. 5). The mean (Eq. 6) of this distribution corresponds to the predicted yield while the variance (Eq. 7) provides the uncertainty associated with the prediction.

### Evaluation of model performance

For evaluation of model performance, the data is randomly split into 70:10:20 train-validation-test sets. The model hyperparameters are tuned on the validation set while the test set is used to evaluate the model performance. The best set of hyperparameters is then used to obtain predictions on the test set. The yield is standardized to have zero mean and one variance over the training data. Ten different train-test splits are considered and the final performance is reported as an average over 10 independent runs. The error is reported as the standard error over these 10 runs. The model predictive performance is measured in terms of root mean squared error (RMSE) and mean absolute error (MAE). The R-squared ($R^2$) value is also computed as a measure of performance for different methods.

**Fig. 2 | An overview of the DKL model architecture with nonlearned input representation.** The reaction features are obtained by concatenating the input representation of individual reaction components. It is then passed through the NN to obtain an embedding which is then used as an input to the GP.

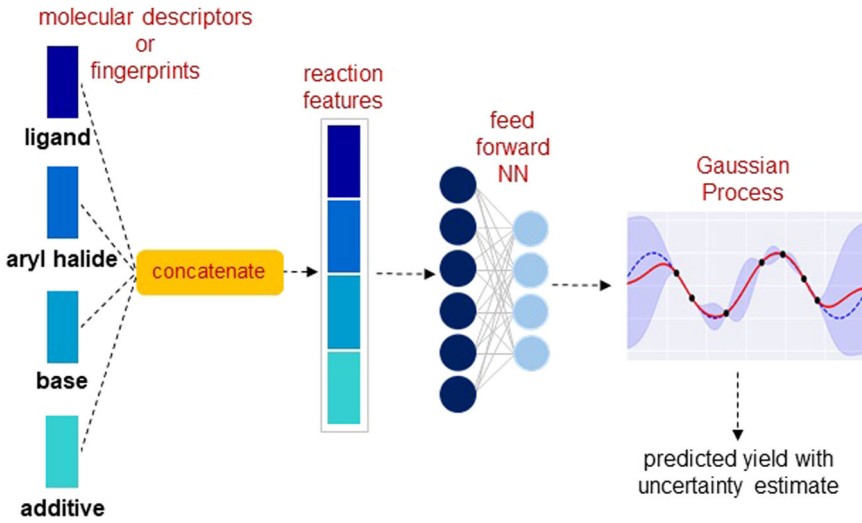

**Fig. 3 | An overview of the DKL model architecture with learned input representation.**
**a** Representative example of graph representation for a molecule with atom and bond features, respectively, shown in green and red colors. **b** A schematic diagram of the DKL model with molecular graph as an input representation. For each reactant, the graph is first passed to the message-passing neural network to learn the graph embedding. The reaction embedding is obtained by summing the graph embedding of individual reaction components. It is then processed by a feed-forward NN and then used as an input to the GP.

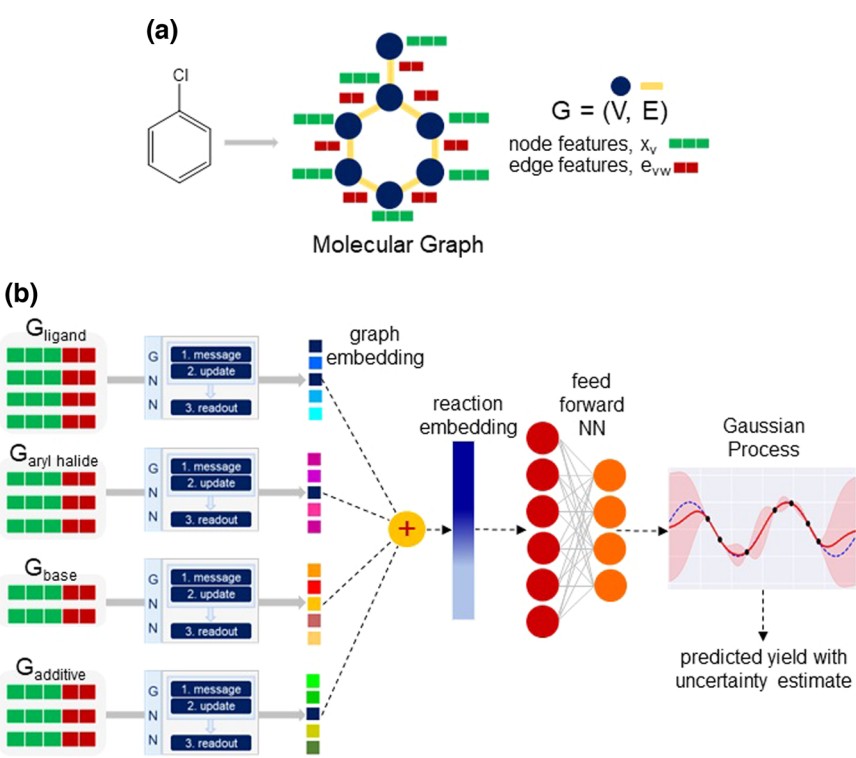

The details of the architecture used for the GNN component of our model are provided in the Supplementary Information (Supplementary Section 3.1). All the calculations are implemented using the PyTorch-based[46] GPyTorch[47] library. A dropout rate of 0.1 is applied to the two fully connected NN layers (Figs. 2b and 3). The embedding thus obtained is passed as an input to the GP, and Matérn52 without automatic relevance determination (ARD)[48] is chosen as the base kernel. The model is trained for 400 epochs. The Adam optimizer[49], with a learning rate of 0.001, is used to update all the parameters of the DKL model.

To show the significance of our method, we compare our model's performance with that of standard Gaussian processes. All the reaction representations used with the DKL model, that is, the molecular descriptors, Morgan fingerprints (with radius 2 and 2048 bits) and DRFP (with 2048 bits), are then used as an input to a standard GP for yield prediction. The GP is trained using the L-BFGS-B optimizer[50] with the Matérn52 kernel without ARD. The L-BFGS-B optimizer is a second-order optimization algorithm.

The second-order derivative (Hessian) can be followed to estimate the minima of the objective more efficiently. It approximates the inverse of the Hessian matrix instead of recalculating it at every iteration. Since the size of Hessian depends on the number of input examples, using L-BFGS-B for large datasets can be computationally expensive.

We obtained an average test RMSE of $8.58 \pm 0.06$ with molecular descriptors (Moldesc-GP). Similar RMSEs of $6.39 \pm 0.06$ and $6.46 \pm 0.05$ are noted, respectively, for Morgan fingerprints (MorganFP-GP) and DRFP (DRFP-GP). Interestingly, a significant gain in performance is noted when the same representations are used with the DKL model. The most significant improvement is with molecular descriptors with an RMSE of $4.87 \pm 0.07$ (Moldesc-DKL). The DKL model with Morgan fingerprints (MorganFP-DKL) and DRFP (DRFP-DKL) also shows improved performance with RMSEs of $4.86 \pm 0.08$ and $4.87 \pm 0.11$, respectively (Fig. 4a). Additionally, we have also investigated the model performance by summing the fingerprints of different reaction components instead of concatenation

**Fig. 4 | Evaluation of model performance.** Summary of results in terms of **a** RMSE (lower the better) and **b** $R^2$ (higher the better) values for GP and DKL model with various molecular representations. **c** A comparison of test set RMSEs for different percentages of the training data. The error bars are the standard error averaged over 10 runs.

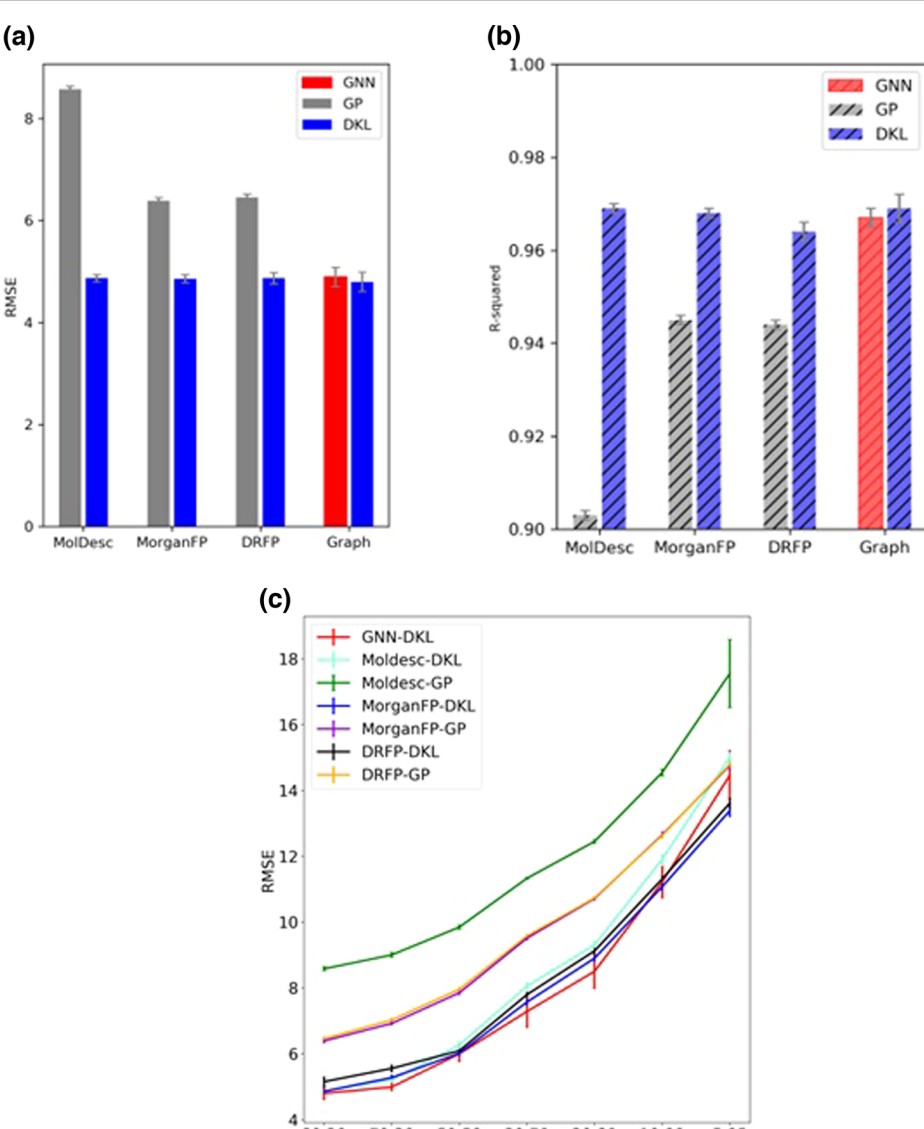

(Supplementary Section 3.2). Finally, we use the DKL model for yield prediction using the graph representation (GNN-DKL) (Fig. 2a). While a significant improvement over standard GPs is noted with an RMSE of 4.80 ± 0.19, the performance is comparable to DKL with other nonlearned representations. The GNNs are known for their state-of-the-art performance in various molecular tasks. Therefore, a stand-alone GNN with an architecture identical to the GNN component of DKL is also used for comparison. With an RMSE of 4.89 ± 0.19, plain GNNs achieve comparable performance to that of DKL (Fig. 4a) (Supplementary Section 4). But unlike GNNs, the DKL model also provides uncertainty estimates. The summary of the results is also presented using the $R$-squared ($R^2$) value as the measure of the performance of different models (Fig. 4b). The $R^2$ plot reveals similar trends as the RMSE plot (Fig. 4a). The $R^2$ value of the DKL model is consistently better than that of standard GPs, whereas it is comparable to that of the GNN.

As mentioned earlier, the GP models are known for their reliable uncertainty estimates. After comparing the predictive performance of a standard GP with the DKL method, we next evaluate their respective uncertainty estimates. The uncertainty estimates are quantified using the negative log predictive density (NLPD). It is a standard probabilistic metric to evaluate the performance of GP models. It computes the negative log-likelihood of the test samples given the predictive

distribution as shown in Eq. (1).

$$\sum_{i=1}^{n} \log p(y_i = t_i | x_i) \qquad (1)$$

The NLPD penalizes both over- and under-confident predictions and, therefore, is suitable for assessing the quality of uncertainty estimates. Since the predictive density can also take values greater than one (unlike probability), the log-likelihood can be both positive and negative. The higher the value of log-likelihood (or lower the value of negative log-likelihood), the better the model fits the observed data. The NLPD values averaged over 10 independent runs for an 80:20 data split are provided in Table 1. It is found that the DKL approach provides better predictive uncertainty estimations compared to standard GPs. The same trend is observed across all molecular representations. We also compared the correlation between absolute prediction error and uncertainty estimates provided by the MorganFP-DKL model and an ensemble of GNNs. The mean and variance of the posterior predictive distribution (Eqs. (6) and (7)) obtained from the MorganFP-DKL model are used as the predicted yield and uncertainty score, respectively. For an ensemble of GNNs, the mean and variance of the predictions across different ensembles are considered respectively for the predicted yield and the uncertainty

estimate. The Spearman rank correlation coefficient ($\rho$) is then computed between absolute error and uncertainty of predicted yields on the test set. We found that the MorganFP-DKL model has a correlation coefficient comparable to that of the GNN ensemble with $\rho = 0.30$. In addition, the model performance is also evaluated on four out-of-sample splits, where the additives present in the training set are absent from the test set and vice versa[51]. The DKL model obtains comparable performance to the previous methods (Supplementary Section 5). The effect of the length of the fingerprint vector is also studied (Supplementary Section 6).

We also investigated the effect of training set size on model performance. The different train-test splits considered for this purpose are 80:20, 70:30, 50:50, 30:70, 20:80, 10:90, and 5:95 (Supplementary Section 7). The hyperparameters are kept the same across different splits. The summary of results is presented in Fig. 4c, which reveals certain interesting insights. The

Moldesc-GP performs worse than other methods across all data splits. On the other hand, the MorganFP-GP and DRFP-GP have similar performance across all splits but are not better than other DKL methods. For the DKL methods, all input representations have a comparable performance when sufficient data is available for training. A slight difference in performance is noted when the amount of training data is small. For instance, with only 5 percent training data, DKL with molecular graph as input (GNN_DKL) has an RMSE similar to a standard GP with fingerprints (DRFP_GP). On the other hand, DKL with fingerprints as input (Morgan_DKL and DRFP_DKL) obtains better performance than standard GPs, even with smaller amounts of training data (Fig. 4c).

## Feature analysis

It can be noted from Fig. 4a that the DKL method with nonlearned representations gives a significant performance improvement over standard GP. To get better insight, we decided to analyze the latent space of the DKL model and compare it with the nonlearned representation space (Supplementary Section 8). Since the Morgan fingerprints consistently provide good predictions across different data sizes, we use them as a representative case. First, we train a DKL model on 80% of the data using Morgan fingerprints. For the 20% test data, the embedding from the last layer of the FFNN (Fig. 3) is extracted. To aid visualization, a UMAP (Uniform Manifold Approximation and Projection) is used for dimensionality reduction[52]. A k-means clustering is performed on the two components of UMAP (Fig. 5a). A similar procedure of dimensionality reduction followed by clustering is performed with the Morgan fingerprints (Fig. 5b). Four distinct clusters could be observed for each case (shown in four different colors). A closer look into the identity of the clusters with Morgan fingerprints (Fig. 5b) revealed that they are based on the similarity of ligand and base. For instance, the blue and red clusters have bases **B1** and **B2**,

**Table 1 | The NLPD values (lower the better) of different models for the 80:20 data split**

| Representation | Method | NLPD |
|---|---|---|
| Graph | GNN | $3.750 \pm 0.006$ |
| Graph | DKL | $-0.209 \pm 0.075$ |
| Molecular descriptor | GP | $0.235 \pm 0.003$ |
| Molecular descriptor | DKL | $-0.275 \pm 0.003$ |
| Morgan FP | GP | $-0.114 \pm 0.003$ |
| Morgan FP | DKL | $-0.289 \pm 0.003$ |
| DRFP | GP | $-0.100 \pm 0.003$ |
| DRFP | DKL | $-0.271 \pm 0.009$ |

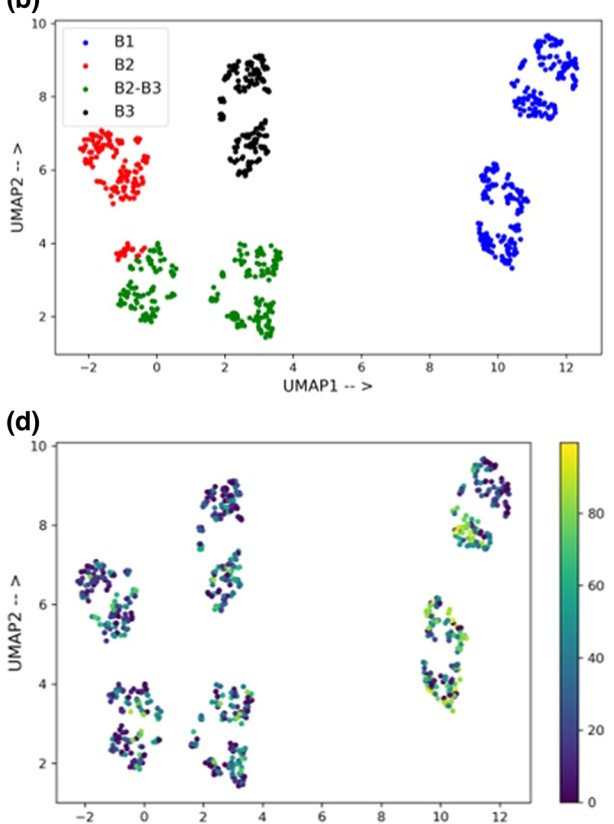

**Fig. 5 | Feature analysis of the embeddings obtained from the DKL model.** UMAP plot for the feature analysis of **a** the latent space of the DKL model and, **b** Morgan fingerprints. The colors represent different clusters identified using

k-means clustering. The legend shows the bases present in the respective clusters, the identity of which is provided in Table S8. The plot shows the distribution of yield in each cluster formed from the **c** DKL features and, **d** Morgan fingerprints.

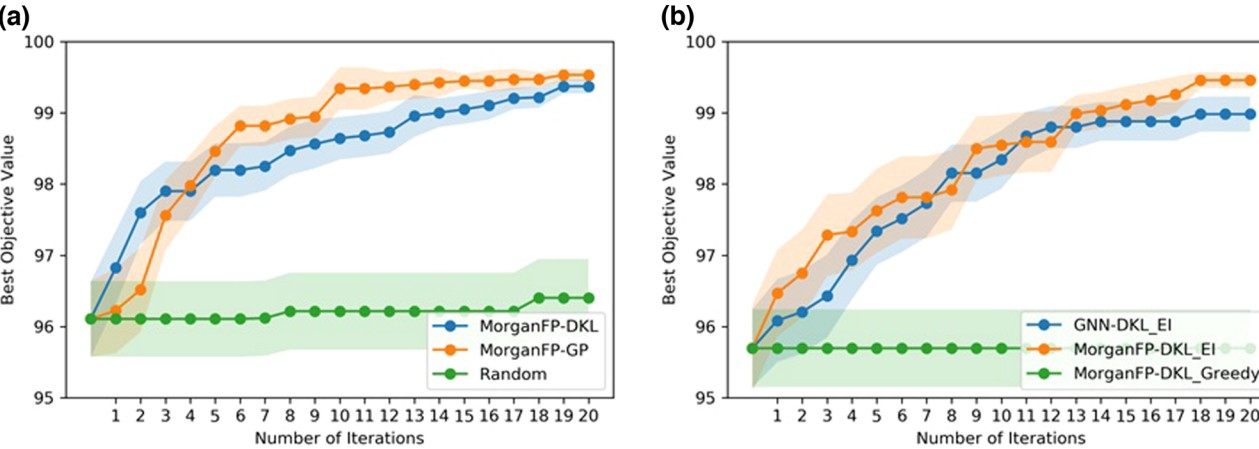

**Fig. 6 | Evaluation of BO performance.** The plot shows the comparison of BO performance for **a** the MorganFP-DKL model and standard MorganFP-GP with random search as the baseline, and **b** the MorganFP-DKL with greedy search and expected improvement as acquisition functions along with the GNN-DKL model.

respectively (see Table S6 for more details), whereas the green cluster has a combination of two bases (**B2**–**B3**). On the other hand, no such conclusion could be drawn from the clusters shown in Fig. 5a. Therefore, the composition of clusters is also analyzed based on their yield distribution. The median yield of the 20% test data is found to be 30.1. The median yield of different clusters in Fig. 5b is 38.4, 21.0, 29.0, and 27.1. These are very close to the median yield of the test data (Fig. 5d). The distribution of yield in each of these clusters is shown in Fig. S4. It can be noted that all four clusters have similar yield distribution. On the other hand, the median yield of clusters formed from the features of MorganFP-DKL model is 21.9, 3.4, 60.6, and 19.0. In this case, the two clusters can clearly be seen as low and high-yield clusters (Fig. 5c). The yield distribution in each cluster is shown in Fig. S5. The three clusters have similar distributions skewed towards low yields, while one cluster has a different distribution with more high-yield samples. This implies that while the Morgan fingerprint contains the structural information, the DKL model built on top of it extracts features that are more suitable for the given task.

Some important points can be gathered from the results discussed so far. Firstly, introducing an NN between the nonlearned input representation and GP can provide extra flexibility to capture more relevant features, leading to improved model performance. It is also noteworthy that the DKL model with nonlearned representation has fewer learnable parameters. This implies that it is both time economic and less resource intensive when compared to training a GNN. Although the predictive performance of our DKL model is similar to GNNs, it additionally provides reliable uncertainty estimates. Additionally, the DKL with nonlearned representations shows an improved performance over GNNs in a low-data regime (Table S3). After establishing that the DKL model provides comparatively better predictions and uncertainty quantification than standard GPs, we further demonstrate its utility as a surrogate model in BO.

**Bayesian optimization using deep Kernel surrogate**

In the past few years, Bayesian optimization has been used in reaction optimization to sequentially suggest new reaction conditions for faster convergence to the target objective[53,54]. A predictive ML model coupled with BO has great potential to accelerate the reaction discovery. To employ BO for reaction optimization, the reaction needs to be represented in a machine-readable format. The nonlearned representations, for instance, physical organic descriptors, molecular fingerprints, etc., are the most commonly used input representations in BO. On the other hand, the use of popular learned representations such as SMILES and molecular graphs is not straightforward. BO is often challenging in problems with high dimensions and non-continuous input space. To address this issue, we combine DKL with BO to automatically identify a low-dimensional subspace. This approach is similar to a standard BO, except using a deep kernel surrogate instead of a standard GP.

In low-data settings, the DKL model gives the best performance with Morgan fingerprints (MorganFP-DKL, Fig. 4c). We performed BO with a deep kernel surrogate using Morgan fingerprint as the input representation. The DKL architecture remains the same as discussed in the previous section (Fig. 3). The BO is performed over the Buchwald-Hartwig dataset for reaction optimization. A general BO loop involves querying reactions from a discrete set of heldout reactions. The acquisition function is evaluated on the heldout set, and the reaction that gets the maximum value is selected as the next candidate at each iteration. Here, we use 5% of the data to initialize the model, while 95% is used as the heldout set. Expected improvement is considered for the acquisition function. The BO is then run for 20 iterations for sequential reaction selection. The BO performance is reported as an average of over 50 randomly initialized trials. To compare the performance of our BO-DKL model, we run BO using a standard GP surrogate on Morgan fingerprints. In addition, a random search is chosen as a baseline. The result is shown in Fig. 6a. It can be noted that while both models outperform random search, the deep kernel method achieves slightly better BO performance than the standard GP. In addition, we optimized the predictive mean of the GP (uncertainty is not used) in the acquisition function. A comparison of BO performance with expected improvement as acquisition function is reported in Fig. 6b as an average over 20 randomly initialized trials. It can be noted from Fig. 6b that the BO performance with the acquisition function that uses uncertainty performs better than the one where uncertainty is not used. In addition, we have also demonstrated the BO with a deep kernel surrogate using a molecular graph as the input representation (Fig. 6b). It can be noted that the BO with MorganFP-DKL performs slightly better than the GNN-DKL model. These results demonstrate that our DKL model can provide good predictive performance along with reliable uncertainty estimates, thereby making it a suitable candidate to be included in the reaction discovery pipeline.

**Conclusion**

We report a deep kernel learning model that combines the representation learning strength of NNs with the uncertainty estimates of GPs for the task of reaction outcome prediction. The ability of our DKL model is demonstrated using the Buchwald-Hartwig cross-coupling with 3955 reactions. For molecular descriptors as input features, the DKL model was able to make accurate predictions with an RMSE of 4.87 ± 0.07. In the case of fingerprints as input features, the model could obtain an RMSE of 4.86 ± 0.08 and 4.87 ± 0.11 with the Morgan fingerprints and DRFP, respectively. A similar RMSE of 4.80 ± 0.19 was obtained with molecular graphs as input features. It is, therefore, reassuring that our DKL model is

applicable to both learned and nonlearned molecular representations as an input. Also, the performance of the DKL model was found to be comparable to the one of graph neural networks, but with uncertainty quantification. Furthermore, the uncertainty estimates provided by the DKL model facilitate its incorporation as a surrogate in Bayesian optimization. It allows the nonlearned molecular representations, such as graphs or high-dimensional fingerprints, to be directly used as an input to BO. Therefore, we believe that our DKL approach will significantly advance the use of GPs with molecular data and has the potential to accelerate reaction discovery by integrating the predictive model with BO.

## Methods

### Deep Kernel learning

A Gaussian process can be viewed as an infinite collection of random variables, any finite number of which follows a joint Gaussian distribution specified by mean m($\cdot$) and positive-definite covariance function k($\cdot, \cdot$) as shown below

$$y(\cdot) \sim GP(m(\cdot) \, k(\cdot, \cdot)). \tag{2}$$

Now, to use the GP technique in practice, we need to choose between various mean and covariance functions in the light of the data. This process is usually referred to as training the GP model. We first choose a GP prior, where the mean and covariance functions are parameterized in terms of hyperparameters. The performance of a GP is greatly influenced by the choice of covariance or *kernel* function $k$. Some popular kernels used with GPs include linear, radial basis function (RBF), periodic, Matern, etc. The kernel $k_\theta(\cdot, \cdot)$ is parameterized by hyperparameters $\theta$, that determine the shape and complexity of the functions in the input space. For instance, the lengthscale hyperparameter of the RBF kernel governs the smoothness of the function, with shorter lengthscales corresponding to a more rapidly varying function with inputs. The inference about all the GP hyperparameters is made using the training data. For this purpose, the probability of data given the hyperparameters is computed, also known as log marginal likelihood $\{\log p(y, |, x, \theta)\}$. Using this, we can find the values of hyperparameters that optimize the log marginal likelihood. Any first-order (such as gradient descent) or second-order (such as L-BFGS-B) optimizers can be used to find the optimal value of hyperparameters. It should also be noted that along with using the kernels individually in a GP model, they can also be combined to get a more specialized kernel. Designing a kernel suitable for a given problem is often tricky.

In deep kernel learning, the kernel is constructed in a manner such that the representational flexibility of NNs is encapsulated in it. More formally, a neural network $g_\phi$, parameterized by $\phi$, transforms the input $x$ and extracts the feature representation $g_\phi(x)$, which is subsequently used as an input to the base kernel $k\left(x_i, x_j | \theta\right)$ with hyperparameter $\theta$ as shown below

$$k_{DKL}\left(\boldsymbol{x_i}, \boldsymbol{x_j} | \theta, \phi\right) = k\left(g_\phi(\boldsymbol{x_i}), g_\phi(\boldsymbol{x_j}) | \theta\right). \tag{3}$$

The deep kernel $k_{DKL}\left(\boldsymbol{x_i}, \boldsymbol{x_j} | \theta, \phi\right)$ provides a scalable closed formed covariance to specify deep kernel GP priors $p(f) = GP(f; 0, k_{DKL}(\cdot, \cdot))$. All parameters $\{\theta, \phi\}$ of the deep kernel are learned jointly by optimizing the log marginal likelihood $\mathcal{L}$ on the training data (X, y), similar to standard GP inference as shown below

$$\mathcal{L} = \log p(\boldsymbol{y} | \theta, \phi, \boldsymbol{X}) \propto -\left[\boldsymbol{y^T}\left(K_{\theta, \phi} + \sigma^2 \mathrm{I}\right)^{-1} \boldsymbol{y} + \log\left|K_{\theta, \phi} + \sigma^2 \mathrm{I}\right|\right]. \tag{4}$$

where $\sigma^2$ is the noise variance. The expression for the log marginal likelihood in Eq. (4) trades off data fit and model complexity. The posterior predictive distribution of the GP evaluated on the test data $\left(X_*, y_*\right)$ is a multivariate Gaussian, characterized by mean and variance values as shown below

$$y_* | X_*, \mathbf{X}, \mathbf{y}, \theta, \phi, \sigma^2 \sim \mathcal{N}(\mathbb{E}[y_*], \mathrm{cov}(y_*)), \tag{5}$$

where,

$$\mathrm{E}[y_*] = \mu_{X_*} + K_{X_*, X}[K_{X, X} + \sigma^2 \mathrm{I}]^{-1} \boldsymbol{y}, \tag{6}$$

$$\mathrm{cov}(y_*) = K_{X_* X_*} - K_{X_*, X}[K_{X, X} + \sigma^2 \mathrm{I}]^{-1} K_{X, X_*}. \tag{7}$$

### Bayesian Optimization

BO is a sequential model-based approach to optimize a black-box objective function that otherwise would be expensive to evaluate[55]. For instance, the number of evaluations can be limited due to the cost associated with purchasing laboratory chemicals. The aim of BO is to solve an optimization problem of the form

$$\boldsymbol{x}^* = arg\,max_{\boldsymbol{x} \in \mathcal{X}} f(\boldsymbol{x}) \tag{8}$$

where f(x) is the objective function over the input domain $\mathcal{X}$. The two key components of BO are a surrogate model and an acquisition function. The surrogate model is used to approximate the objective function. A GP is usually the most popular choice, owing to its ability to provide calibrated uncertainty estimates. The acquisition function then utilizes the uncertainty estimates to propose the next sampling point such that there is a compromise between exploitation and exploration. Examples of acquisition functions include expected improvement, upper confidence bound, entropy search, Thompson sampling, etc. The new sample is added to previous samples, and the posterior of the surrogate model is updated. This process is continued iteratively until convergence or budget depletion.

In this study, we use expected improvement (EI) as the acquisition function. It computes the expectation of the improvement to select a sample f(x) that, in expectation, improves the most upon the best sample f(x$^+$) observed so far, as shown below

$$\mathrm{EI}(\boldsymbol{x}) = \mathbb{E}\left(f(\boldsymbol{x}) - f(\boldsymbol{x}^+), 0\right) \tag{9}$$

This is computed for every point, and the one providing a greater EI is selected. In case of no improvement, EI is zero.

## Data availability

The data is publicly available through GitHub. Additional figures, tables and technical details are provided in the Supplementary Information.

## Code availability

The code is publicly available through GitHub.

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

## Acknowledgements
J.M.H. and S.S. acknowledge support from a Turing AI Fellowship under grant EP/V023756/1.

## Author contributions
S.S.: conceptualization, investigation, methodology, analysis, writing the original draft. J.M.H.: analysis, writing-review, editing, supervision.

## Competing interests
The authors declare no competing interests.
