## [Peer Review File · Communications Chemistry]

Reviewers' comments:

Reviewer #1 (Remarks to the Author):

A deep kernel learning (DKL) approach is applied in a chemistry setting. It is not clear if this is the first application of DKL to a chemistry problem or not. The authors might explicitly address this in the manuscript.

A comparable performance to graph neural networks is reported on one (well-studied literature) dataset. The advantage that the approach offers is that it provides an uncertainty estimate.

Major points

The abstract might benefit from presentation of a quantitative finding.

Page 6. The reaction yield ranges from 0 to ~100% (units should be included). It might be noted that the distribution of yields is far from uniform. Perhaps there are consequences of this non-uniform distribution that should be considered.

Page 10. The phrase "to ensure technical rigor" is a bit vague. Perhaps the issue could be stated more explicitly? And why 10 repeats and not, say, 3 or 100?

Page 11. Should the discussion of relative performance include some statistical significance tests?

Page 13. The "feature analysis" focuses on the presence or absence of certain chemical species. This would seem to be in the realm of "one-hot encoding" (as considered as a baseline in ref #31). So this is not really directly addressing chemical features. More generally, the study does not provide any new chemical insight.

Page 14. The statement that "the DKL model ... extracts features which are more suitable" is rather vague and not fully evidenced.

Page 15. Is there a concrete example illustrating the benefit of "reliable uncertainty estimates" in this setting? Is there any notable variation in the uncertainty of predictions across the dataset? The construction of the dataset in ref #31 was motivated in part by the desire to reduce that variation.

Page 16. The starting point of the yield optimisation appears in Fig 6 to be 96%. Is an increase of a few % of importance? Ref #31 constructed the dataset with the deliberate introduction of additives (one typically encountered in med chem) that were inhibitory, i.e., suppressed yield. Is this dataset then the most suitable/typical for yield optimisation? Removing an inhibitory additive seems like a very straightforward optimisation.

SI. Code is (rather will be) available.

Minor points

Page 2. "utility values" is unclear

Page 12. A sentence should not begin with "Whereas"

Page 17 "assuring" should be reassuring"

Reviewer #2 (Remarks to the Author):

The paper introduces an approach where neural networks are combined with Gaussian processes (GPs) to allow for reaction outcome prediction with uncertainty. This is then used as input to Bayesian optimization for reaction optimization. The presented approach is very interesting, and uncertainty prediction is valuable for many areas, not only reaction outcome prediction. Although the project is well formulated, it does not appear complete in its current shape, and the manuscript needs further work. I have listed a few points which should be addressed before it can be accepted for publishing:

Introduction / Methods

* The way the introduction is written, it seems like the final model will be the GNN-DKL, but in fact it is the Morgan FP-DKL. To avoid confusion for the reader, I suggest switching the order of how these are presented, add them under a joint section, or shift the focus in some other way. Fig. 2 and 3 could be merged.

* The explanations of GPs could be improved for readers who are not familiar with these models:

- $K\theta, \phi$ is not defined properly, the relationship between f and y is not explicitly stated (e.g. in Eqs. 4-6 where f^* is used for evaluating (X^*, y^*))

- Explain the difference between (X^*, y^*) and (X, y) . Might be even better to use (X, y) and $(X_{\text{train}}, y_{\text{train}})$ instead.

* Fig. 2b may be misleading since the same colors are used in the component embeddings as the (summed) reaction embedding. This way, it seems like the components have been concatenated. A suggestion is to use gray-scale.

* Why are the GNN-embeddings summed while the fingerprints are concatenated for the different reaction components prior to inputting to the feedforward network?

* "... the final performance is reported as average over 10 independent runs." There is also an error reported (e.g. 5 ± 0.01). Is this the standard deviation over those 10 independent runs?

* The data is according to the Methods section split into 70:10:20, but in the Results, the authors report results for splits 80:20, 70:30. It would be helpful to describe what happens with the 70:20:10 split, the training/validation procedure, as well as how the 80:20, etc. testing is done.

Results

* Table S1 is useful and could be added to the main text.

* Fig. 4: Use the same model colors for subplots a) and b) to denote the models. The legend labels should be updated to the model names, removing underscores. The train-test split analysis could be made clearer by adding the errorbar above the markers (different color than the markerfacecolor. Maybe add caps to the errorbars as done in subplot a). It would also be nice to have the R^2 for each model (reported in Table S1) in a subplot.

* What does an RMSE of e.g. 6 entail about performance? It would be helpful to compare the yield to not just the GNN that the authors developed, but also to state-of-the-art like gradient-boost, random

forest, or yield-BERT (see <https://doi.org/10.1021/acs.jcim.3c01524> for a recent review)

* The results reported in Supplementary Material are not ordered as they appear in the text.

* Regarding the feature analysis:

- How do you choose the number of clusters (4)? When looking at the data, it looks like more clusters are present for both approaches. The "optimal" number of clusters could be selected with a silhouette score, for example.

- To enhance the discussion about yield-distinction between clusters, you could add a plot with the actual yield distributions for each cluster (e.g. box plots or probability densities), as it is difficult to assess the spread and overlap with the UMAP projections. It makes sense that the latent space encodings separate according to yield since they are trained to encode the data for yield prediction.

Other

* Instead of referring to each model as DKL in the text, you could refer to them as GNN-DKL, MolDesc-DKL, MorganFP-DKL, etc. to be more specific.

Reviewer #3 (Remarks to the Author):

This paper introduces an innovative approach by incorporating deep kernel learning into reaction outcome prediction, offering accuracy comparable to existing methods with the added benefit of providing natural uncertainty measurements. This conceptual advance introduces a novel dimension to the field of reaction modeling and reaction optimization, potentially enhancing experimental design processes. However, concerns regarding the dataset's choice and clarity in certain methodological explanations warrant attention. Overall, I recommend acceptance after minor revisions addressing these points.

Points to address:

1. Dataset Validity: The exclusive use of the C-N coupling dataset, known for its experimental design flaws [1], raises questions about the model's learning efficacy. Incorporating additional datasets could reinforce the model's validity.
2. Clarification on Model "Training": The use of the L-BFGS-B optimizer for training the Gaussian Process, described on page 11, needs elaboration. Given the non-parametric nature of Gaussian Processes, further explanation of the optimizer's role and its implementation would enhance understanding.
3. Uncertainty Quantification: Despite the emphasis on uncertainty as a critical feature, its discussion and analysis are limited. Expanding on the methodology, specifically regarding NLPD, and incorporating these findings into the main text would provide a clearer picture of the model's capabilities.
4. Adding Graph DKL in Bayesian Optimization: An exploration of the Graph DKL model within the Bayesian Optimization setting, complementing the current Figure 6, could offer valuable insights into its performance and applicability in practical scenarios.

Reference:

[1] Chuang, Kangway V., and Michael J. Keiser. "Comment on "Predicting reaction performance in C-N cross-coupling using machine learning"." Science 362.6416 (2018): eaat8603.

Reviewer #4 (Remarks to the Author):

I co-reviewed this manuscript with one of the reviewers who provided the listed reports. This is part of a Communications Chemistry initiative to facilitate training in peer review and to provide appropriate recognition for Early Career Researchers who co-review manuscripts.

Reviewer: 1

R1-C1 A deep kernel learning (DKL) approach is applied in a chemistry setting. It is not clear if this is the first application of DKL to a chemistry problem or not. The authors might explicitly address this in the manuscript. A comparable performance to graph neural networks is reported on one (well-studied literature) dataset. The advantage that the approach offers is that it provides an uncertainty estimate.

We thank the reviewer for the careful assessment of our work. This study is one of the earliest examples of the application of DKL for a chemistry task. We have now added this in the revised manuscript (page 3).

R1-C2 Major points: The abstract might benefit from presentation of a quantitative finding.

As suggested by the reviewer, we have now included the main findings quantitatively in abstract of the revised manuscript.

R1-C3 Page 6. The reaction yield ranges from 0 to ~100% (units should be included). It might be noted that the distribution of yields is far from uniform. Perhaps there are consequences of this non-uniform distribution that should be considered.

As noted by the reviewer, the distribution of yield is not uniform and is skewed towards high yield reactions. This could bias the model to predict only high values. We calculated the RMSE when all the samples are predicted a high value of 80. But the model performance is found to be far worse than that obtained from our trained DKL model. This shows that the model is able to learn reasonably well even with non-uniform distribution.

R1-C4 Page 10. The phrase “to ensure technical rigor” is a bit vague. Perhaps the issue could be stated more explicitly? And why 10 repeats and not, say, 3 or 100?

We have now removed the phrase in the revised manuscript. Ten repeats achieves a trade-off between gaining statistical significance for the results obtained (less affected by random outcomes in split of data or training) and the required computational cost. Also, the standard

deviation of ten repeats across different methods is not very different and allows for a reasonable comparison of model performance.

R1-C5 Page 11. Should the discussion of relative performance include some statistical significance tests?

We have performed paired t-test to analyze the performance of different models. The results are presented in the revised SI (Table S4). It is found that the MorganFP-DKL model performs significantly better than other models, but comparable to GNN-DKL.

R1-C6 Page 13. The “feature analysis” focuses on the presence or absence of certain chemical species. This would seem to be in the realm of “one-hot encoding” (as considered as a baseline in ref #31). So this is not really directly addressing chemical features. More generally, the study does not provide any new chemical insight.

We wish to clarify that for feature analysis, DKL model trained on 80 percent of the data is used. The embedding from the last layer of the FFNN is extracted for the 20 percent test data. For visualization, UMAP followed by k-means clustering is performed. A similar procedure of dimensionality reduction followed by clustering is performed with the Morgan fingerprints. Distinct clusters are identified in both cases and analysis is done based on identity of reactions and their yield distribution in different clusters (section 8 in SI). Therefore, our analysis has nothing similar to one-hot encoding. The focus of feature analysis is to demonstrate that adding a neural network layer to the nonlearned input representation provides flexibility to extract more relevant features and thus improves performance.

R1-C7 Page 14. The statement that “the DKL model ... extracts features which are more suitable” is rather vague and not fully evidenced.

We apologize for not being clear. We have shown in Fig. 5 the UMAP plot obtained from the embedding of the DKL model and Morgan fingerprints. The embeddings of the DKL model

are clustered based on yield of the reaction. This makes it easier for the GPs to make prediction, which is then reflected in an improved performance of DKL model.

R1-C8 Page 15. Is there a concrete example illustrating the benefit of “reliable uncertainty estimates” in this setting? Is there any notable variation in the uncertainty of predictions across the dataset? The construction of the dataset in ref #31 was motivated in part by the desire to reduce that variation.

The benefit of uncertainty estimates provided by the deep kernel learning model is illustrated by using it for Bayesian optimization to optimize the reaction outcome (section: Bayesian Optimization using Deep Kernel Surrogate). A comparison of BO performance when uncertainty is not used is included in Fig. S6(b) in the revised SI. The BO performance where uncertainty estimates are used is found to be better than the case where simply the predictive mean of GP is optimized without using uncertainties. The uncertainty estimates are quantified using the negative log predictive density (NLPD) metric (Table S2). It can be noted that the standard deviation across different test sets is very less.

R1-C9 Page 16. The starting point of the yield optimisation appears in Fig 6 to be 96%. Is an increase of a few % of importance? Ref #31 constructed the dataset with the deliberate introduction of additives (one typically encountered in med chem) that were inhibitory, i.e., suppressed yield. Is this dataset then the most suitable/typical for yield optimisation? Removing an inhibitory additive seems like a very straightforward optimisation.

SI. Code is (rather will be) available.

The data used in this work is a well-studied dataset in literature. We agree with the reviewer that this dataset might not be the most suitable for yield optimization. But the primary purpose here is to demonstrate that the uncertainty estimates provided by the DKL method can be used to optimize reaction outcome in a Bayesian optimization setup. We found that BO performance

with uncertainties obtained by DKL method is better as compared to when the uncertainty estimates are not used (Fig. S6(b)).

R1-C10 Minor points: Page 2. “utility values” is unclear

This is now changed in the revised manuscript.

R1-C11 Page 12. A sentence should not begin with “Whereas”

This is now changed in the revised manuscript.

R1-C12 Page 17 “assuring” should be reassuring”

This is now changed in the revised manuscript.

Reviewer: 2

R2-C1 The paper introduces an approach where neural networks are combined with Gaussian processes (GPs) to allow for reaction outcome prediction with uncertainty. This is then used as input to Bayesian optimization for reaction optimization. The presented approach is very interesting, and uncertainty prediction is valuable for many areas, not only reaction outcome prediction. Although the project is well formulated, it does not appear complete in its current shape, and the manuscript needs further work. I have listed a few points which should be addressed before it can be accepted for publishing:

We thank the reviewer for appreciating our work. We have now added the required details in the revised manuscript and SI.

R2-C2 Introduction / Methods

* The way the introduction is written, it seems like the final model will be the GNN-DKL, but in fact it is the Morgan FP-DKL. To avoid confusion for the reader, I suggest switching the order of how these are presented, add them under a joint section, or shift the focus in some other way. Fig. 2 and 3 could be merged.

As suggested by the reviewer, we have changed the order of sections. DKL with nonlearned molecular representation is presented first followed by DKL with learned molecular representation.

R2-C3 * The explanations of GPs could be improved for readers who are not familiar with these models:

- $K_{\theta, \phi}$ is not defined properly, the relationship between f and y is not explicitly stated (e.g. in Eqs. 4-6 where f^* is used for evaluating (X^*, y^*))
- Explain the difference between (X^*, y^*) and (X, y) . Might be even better to use (X, y) and $(X_{\text{train}}, y_{\text{train}})$ instead.

As suggested by the reviewer, we have made changes in the revised manuscript.

R2-C4 * Fig. 2b may be misleading since the same colors are used in the component embeddings as the (summed) reaction embedding. This way, it seems like the components have been concatenated. A suggestion is to use gray-scale.

We have now changed the colors of the reaction embedding in the revised manuscript.

R2-C5 * Why are the GNN-embeddings summed while the fingerprints are concatenated for the different reaction components prior to inputting to the feedforward network?

The model performance obtained by summing the fingerprint embeddings similar to graph embeddings is included in the revised supporting information (Figure S2 and Table S1). The results obtained are found to be comparable to that from concatenating the fingerprint of different reaction components.

R2-C6 * "... the final performance is reported as average over 10 independent runs." There is also an error reported (e.g. 5 ± 0.01). Is this the standard deviation over those 10 independent runs?

The error reported is the standard deviation over 10 independent runs.

R2-C7 * The data is according to the Methods section split into 70:10:20, but in the Results, the authors report results for splits 80:20, 70:30. It would be helpful to describe what happens with the 70:20:10 split, the training/validation procedure, as well as how the 80:20, etc. testing is done.

The data is first partitioned into 70:10:20 train-validation-test split. The validation set is used to tune the hyperparameters. The best set of hyperparameters thus obtained are used for predictions on the test set. When using other splits such as 70:30 or 50:50, the same values of hyperparameters are used. We have now clarified this in the revised manuscript (pages 10 and 13).

R2-C8 Results

* Table S1 is useful and could be added to the main text.

The data in Table S1 is presented as a bar plot in Figure 4a of the main text for an easier comparison of the model performance.

R2-C9 * Fig. 4: Use the same model colors for subplots a) and b) to denote the models. The legend labels should be updated to the model names, removing underscores. The train-test split analysis could be made clearer by adding the errorbar above the markers (different color than the markerfacecolor. Maybe add caps to the errorbars as done in subplot a). It would also be nice to have the R² for each model (reported in Table S1) in a subplot.

The plot in Fig. 4a is for the comparison of DKL model with standard GP. We have used only two colors for better clarity. Using same model colors as Fig. 4b does not help in conveying the message. So, we wish to retain the same colors in Fig. 4b.

As suggested by the reviewer, we have made modifications to Fig. 4b and added R² plot for each model in the revised manuscript.

R2-C10 * What does an RMSE of e.g. 6 entail about performance? It would be helpful to compare the yield to not just the GNN that the authors developed, but also to state-of-the-art like gradient-boost, random forest, or yield-BERT (see <https://doi.org/10.1021/acs.jcim.3c01524> for a recent review)

The data used in this work is a well-studied dataset in literature. In previous work, GNNs have been shown to provide state-of-the-art performance (ref. 16). Thus, we have compared our DKL model performance with that of GNNs. We have now cited the review in the revised manuscript (ref. 11).

R2-C11 * The results reported in Supplementary Material are not ordered as they appear in the text.

As suggested by the reviewer, we have now ordered the results in SI as they appear in the main text.

R2-C12 * Regarding the feature analysis:

- How do you choose the number of clusters (4)? When looking at the data, it looks like more clusters are present for both approaches. The “optimal” number of clusters could be selected with a silhouette score, for example.

We used the elbow method to determine the optimal number of clusters. The details are now included in the revised SI (Fig. S3).

R2-C13 - To enhance the discussion about yield-distinction between clusters, you could add a plot with the actual yield distributions for each cluster (e.g. box plots or probability densities), as it is difficult to assess the spread and overlap with the UMAP projections. It makes sense that the latent space encodings separate according to yield since they are trained to encode the data for yield prediction.

We have now included the distribution of yield for each cluster in the revised SI (Figs. S4 and S5).

R2-C14 Other

* Instead of referring to each model as DKL in the text, you could refer to them as GNN-DKL, MolDesc-DKL, MorganFP-DKL, etc. to be more specific.

We have now modified this in the revised manuscript.

Reviewer: 3

R3-C1 This paper introduces an innovative approach by incorporating deep kernel learning into reaction outcome prediction, offering accuracy comparable to existing methods with the added benefit of providing natural uncertainty measurements. This conceptual advance introduces a novel dimension to the field of reaction modeling and reaction optimization, potentially enhancing experimental design processes. However, concerns regarding the dataset's choice and clarity in certain methodological explanations warrant attention. Overall, I recommend acceptance after minor revisions addressing these points.

We thank the reviewer for appreciating our work. We have now added the required details in the revised manuscript and SI.

R3-C2 Points to address:

1. Dataset Validity: The exclusive use of the C-N coupling dataset, known for its experimental design flaws [1], raises questions about the model's learning efficacy. Incorporating additional datasets could reinforce the model's validity.

Due to the known concerns of dataset as pointed out by the reviewer, we have also performed calculations on more challenging out-of-sample splits (ref. 51). The results are presented in the supporting information (section 5).

R3-C3 2. Clarification on Model "Training": The use of the L-BFGS-B optimizer for training the Gaussian Process, described on page 11, needs elaboration. Given the non-parametric nature of Gaussian Processes, further explanation of the optimizer's role and its implementation would enhance understanding.

As suggested by the reviewer, additional explanation regarding the L-BFGS-B optimizer is now included in the revised SI (section 3.2).

R3-C4 3. Uncertainty Quantification: Despite the emphasis on uncertainty as a critical feature, its discussion and analysis are limited. Expanding on the methodology, specifically regarding

NLPD, and incorporating these findings into the main text would provide a clearer picture of the model's capabilities.

The utility of uncertainty estimates obtained from the DKL model is shown by using it with Bayesian optimization (section: Bayesian Optimization using Deep Kernel Surrogate). A comparison of BO performance when uncertainty is not used is included in Fig. S6(b) in the revised SI. The BO performance where uncertainty estimates are used is found to be better than the case where simply the predictive mean of GP is optimized without using uncertainties. We have included additional text in the revised SI (section 4) explaining NLPD in more details.

R3-C5 4. Adding Graph DKL in Bayesian Optimization: An exploration of the Graph DKL model within the Bayesian Optimization setting, complementing the current Figure 6, could offer valuable insights into its performance and applicability in practical scenarios.

As suggested by the reviewer, we have now performed Bayesian optimization using the GNN_DKL model and results are included in the revised SI (Figure S6).

Reviewers' comments:

Reviewer #1 (Remarks to the Author):

The authors have considered the comments of the referees and have made a number of changes accordingly. The revised manuscript is now acceptable.

Reviewer #2 (Remarks to the Author):

I believe the authors have done a fine job with incorporating the feedback from the reviewers.

I have only a few final remarks:

* Instead of just writing "This is one of the earliest examples of the application of DKL for a chemistry task" in the introduction, the authors ought to add references to previous work

* In "Deep Kernel Learning with Nonlearned Molecular Representations": "But they typically fail to provide better performance than the learned representations when sufficient data is available." This statement needs a reference, and might be better suited in the "learned representation" section.

* Figure 2 and 3 use different notation for the aryl halide, in Figure 3 the label is unnecessarily abbreviated.

* From prev. review: "... the final performance is reported as average over 10 independent runs." There is also an error reported (e.g. 5 ± 0.01). Is this the standard deviation over those 10 independent runs?

Answer from authors: The error reported is the standard deviation over 10 independent runs.

Comment: This should also be mentioned in the text.

* Thanks for adding a plot of R-squared. A sentence with the conclusion from this plot would be helpful.

* Fig. 4c is improved, but it would be easier to see the plots if the linewidth was increased. The colors are not optimal (the green and yellow are difficult to see, and green and red might be difficult to separate for some people).

* The elbow method is quite subjective and, in this case, does not give a clear elbow. I would recommend using the silhouette score or similar quantitative metric alone or in addition to the elbow plot. Moreover, please state which representation was used to determine the number of clusters. The number of clusters are likely different depending on if DKL latent space or Morgan Fingerprints were used.

* A discussion about the cluster yield distributions (shapes e.g.) is missing in the main text.

* The legend in Figure 5b is slightly confusing as it does come from additional analysis not from the clustering - for a reader just glancing over the figures, it would be natural to assume that the legend also applies to Figure 5a, which it does not. This could be clarified in the figure text.

Reviewer #3 (Remarks to the Author):

Thank the authors for trying to address my concerns. However, all points I've raised are significant and should be addressed directly in the main text, rather than being relegated to the SI. A published paper's main text should contain sufficient detail to fully convey the core message, value, and validity of the proposed model. The current draft, with its minimal revisions to the main text and extensive reliance on the SI, fails to meet this standard and does not adequately address my concerns.

Furthermore, beyond the performance of Bayesian Optimization (BO), a more detailed assessment of the model's uncertainty is warranted. This should include an exploration of its correlation with both the model's error and its distance from the training set. Comparing these findings against more baselines, such as the uncertainty derived from ensemble models, would also be beneficial.

Reviewer #4 (Remarks to the Author):

I co-reviewed this manuscript with one of the reviewers who provided the listed reports. This is part of a Communications Chemistry initiative to facilitate training in peer review and to provide appropriate recognition for Early Career Researchers who co-review manuscripts.

Reviewer: 2

R2-C1 I believe the authors have done a fine job with incorporating the feedback from the reviewers.

We thank the reviewer for acknowledging our response.

R2-C2 I have only a few final remarks:

* Instead of just writing "This is one of the earliest examples of the application of DKL for a chemistry task" in the introduction, the authors ought to add references to previous work

We have included the reference in the revised manuscript (ref. 30).

R2-C3 * In "Deep Kernel Learning with Nonlearned Molecular Representations": "But they typically fail to provide better performance than the learned representations when sufficient data is available." This statement needs a reference, and might be better suited in the "learned representation" section.

We have included the reference in the revised manuscript (refs. 17 and 18).

R2-C4 * Figure 2 and 3 use different notation for the aryl halide, in Figure 3 the label is unnecessarily abbreviated.

We have made the label for aryl halide consistent in the revised manuscript.

R2-C5 * From prev. review: "... the final performance is reported as average over 10 independent runs." There is also an error reported (e.g. 5 ± 0.01). Is this the standard deviation over those 10 independent runs?

Answer from authors: The error reported is the standard deviation over 10 independent runs.

Comment: This should also be mentioned in the text.

We have included the text in the revised manuscript (page 10).

R2-C6 * Thanks for adding a plot of R-squared. A sentence with the conclusion from this plot would be helpful.

We have included the additional text in the revised manuscript (page 12).

R2-C7 * Fig. 4c is improved, but it would be easier to see the plots if the linewidth was increased. The colors are not optimal (the green and yellow are difficult to see, and green and red might be difficult to separate for some people).

We have increased the linewidth and changed colors of Fig. 4c in the revised manuscript.

R2-C8 * The elbow method is quite subjective and, in this case, does not give a clear elbow. I would recommend using the silhouette score or similar quantitative metric alone or in addition to the elbow plot. Moreover, please state which representation was used to determine the number of clusters. The number of clusters are likely different depending on if DKL latent space or Morgan Fingerprints were used.

As suggested by the reviewer, we have performed silhouette analysis and the results are included in the revised SI (section 8).

R2-C9 * A discussion about the cluster yield distributions (shapes e.g.) is missing in the main text.

We have included the discussion on cluster yield distributions in the revised manuscript (page 15).

R2-C10 * The legend in Figure 5b is slightly confusing as it does not come from additional analysis not from the clustering - for a reader just glancing over the figures, it would be natural to assume that the legend also applies to Figure 5a, which it does not. This could be clarified in the figure text.

We have clarified the figure text in the revised manuscript.

Reviewer: 3

R3-C1 Thank the authors for trying to address my concerns. However, all points I've raised are significant and should be addressed directly in the main text, rather than being relegated to the SI. A published paper's main text should contain sufficient detail to fully convey the core message, value, and validity of the proposed model. The current draft, with its minimal revisions to the main text and extensive reliance on the SI, fails to meet this standard and does not adequately address my concerns.

As suggested by the reviewer, we have moved the changes made in the last revision to the main text (pages 11, 13, 14, 17, and 18).

R3-C2 Furthermore, beyond the performance of Bayesian Optimization (BO), a more detailed assessment of the model's uncertainty is warranted. This should include an exploration of its correlation with both the model's error and its distance from the training set. Comparing these findings against more baselines, such as the uncertainty derived from ensemble models, would also be beneficial.

As suggested by the reviewer, we have now added additional details comparing correlation between model's error and uncertainty estimates of DKL model and an ensemble of GNNs in the revised manuscript (page 13). We also wish to clarify that the following experiments have been performed to assess the model's uncertainty estimates:

(1) The quality of uncertainty estimates is evaluated using the negative log predictive density (NLPD) values. The comparison is done with all the models considered in this study, including GPs (which are known to provide reliable uncertainty estimates) and GNNs (which provide comparable performance to DKL). This is now included in the revised manuscript (page 13, Table 1).

(2) The uncertainty estimates are utilized in Bayesian optimization (BO). The BO performance of the DKL model is found to be better than the standard GPs and random search baseline.

(3) The benefit of uncertainty estimates is evaluated by optimizing the predictive mean of the GP (uncertainty is not used) in the acquisition function. We noted that the BO performance with acquisition function that uses uncertainty performs better than the one where uncertainty is not used. This is now included in the revised manuscript (page 18, Fig. 6b).

Reviewers' comments:

Reviewer #2 (Remarks to the Author):

I believe the authors have made the necessary alterations to the manuscript. It is now ready for publication.

Reviewer #3 (Remarks to the Author):

Thanks authors for moving the previous revisions to the main text. After reviewing the modified draft, I found my major concerns were not addressed. The overall quality of the draft has not been significantly improved from the initial version. I keep my comments that this draft can only be published (in any venue) after a revision that could address the concerns.

1. I pointed out that the C-N coupling dataset is well-known for its experimental design flow and suggested testing the proposed method on another dataset. I have not found the new experiments in the revised version. If I missed them, could you point me to them? If not, could you defend the choice of this specific dataset?

2. I pointed out that GP itself is a non-parametric method and doesn't involve an optimization process for "training." So, the sentence "The GP is trained using the L-BFGS-B optimizer" on page 11 is not even correct. I suggested that the way the L-BFGS-B optimizer is used must be explained. The revision added a sentence introducing L-BFGS-B's advantage compared to first-order optimizers, which doesn't help explain how it is used.

3. I thank the authors for adding more details of NLPD. However, I am more confused. A typical NLPD definition is the negative value of the current equation (1), and the smaller, the better in a typical supervised learning manner, which is opposite to the good direction defined in this draft. And given all probability values should be in the range between $[0,1]$, the values should be all negative (based on the current definition from equation 1) or all positive (based on typical NLPD definition). The current reported value of NLPDs has both negatives and positives at the same time, which implies there might be some mistakes in the calculation. Besides, NLPD is only a way to measure if the model is confident in the true label on the test set, while a good model uncertainty should be large in data points far from the training set and small in data points close to the training set. The authors conducted a simple random 80:20 dataset split, which cannot tell what the model should be confident of. I suggested that authors add the correlation between uncertainty and prediction error as an auxiliary metric; however, I can only find one sentence on page 14 that says the Spearson correlation is 0.3, without any details telling what experiments were done and how this value compared to other uncertainty metrics, such as ensemble variance.

4. I asked the authors to add Graph DKL BO to Figure 6, but I can't find it.

Reviewer: 3

R3-C1 Thanks authors for moving the previous revisions to the main text. After reviewing the modified draft, I found my major concerns were not addressed. The overall quality of the draft has not been significantly improved from the initial version. I keep my comments that this draft can only be published (in any venue) after a revision that could address the concerns.

We have now included additional clarifications in the revised manuscript. We hope that with these modifications the reviewer would find our manuscript suitable for publication.

R3-C2 1. I pointed out that the C-N coupling dataset is well-known for its experimental design flow and suggested testing the proposed method on another dataset. I have not found the new experiments in the revised version. If I missed them, could you point me to them? If not, could you defend the choice of this specific dataset?

The reaction considered in this study is of high contemporary importance and finds broad applicability in pharmaceutical synthesis. The Buchwald-Hartwig cross-coupling dataset has been used in multiple studies to evaluate the relative performance of various ML methods. Therefore, it provides a suitable high-throughput experimentation (HTE) dataset to benchmark new predictive ML models. In our previous response, we acknowledged the known concerns of the dataset and considered more challenging splits whereby certain additives are held out and not included in the training set (as suggested by previous work, ref. 52). The model performance on the out-of-sample splits is presented in the supporting information (section 5).

R3-C3 2. I pointed out that GP itself is a non-parametric method and doesn't involve an optimization process for "training." So, the sentence "The GP is trained using the L-BFGS-B optimizer" on page 11 is not even correct. I suggested that the way the L-BFGS-B optimizer is

used must be explained. The revision added a sentence introducing L-BFGS-B's advantage compared to first-order optimizers, which doesn't help explain how it is used.

We wish to clarify that training the GP involves the tuning of hyperparameters (such as lengthscale) by optimizing the log marginal likelihood. For this purpose, the L-BFGS-B optimizer is used to obtain the optimal values of these hyperparameters. A detailed response is as follows:

A Gaussian process can be viewed as an infinite collection of random variables, any finite number of which follows a joint Gaussian distribution specified by mean $m(\cdot)$ and positive-definite covariance function $k(\cdot, \cdot)$ as shown below

$$y(\cdot) \sim GP(m(\cdot)k(\cdot, \cdot)).$$

Now, to use the GP technique in practice, we need to choose between various mean and covariance functions in the light of the data. This process is usually referred to as training the GP model. We first choose a GP prior where the mean and covariance functions are parameterized in terms of hyperparameters. The performance of a GP is greatly influenced by the choice of covariance or *kernel* function k . Some popular kernels used with GPs include linear, radial basis function (RBF), periodic, Matern etc. The kernel $k_\theta(\cdot, \cdot)$ is parameterized by hyperparameters θ , that determine the shape and complexity of the functions in the input space. For instance, the lengthscale hyperparameter of the RBF kernel governs the smoothness of the function, with shorter lengthscales corresponding to a more rapidly varying function with inputs. The inference about all the GP hyperparameters is made using the training data. For this purpose, the probability of data given the hyperparameters is computed, known as *log marginal likelihood* $\{\log p(y|x, \theta)\}$. Using this we can find the values of hyperparameters that optimize the marginal likelihood. Any first-order (such as gradient descent) or second-order (such as L-BFGS-B) optimizers can be used to find the optimal value of hyperparameters. The additional details are now included in the Methods section (pages 3 and 4) in the revised manuscript.

R3-C4 3. I thank the authors for adding more details of NLPD. However, I am more confused. A typical NLPD definition is the negative value of the current equation (1), and the smaller, the better in a typical supervised learning manner, which is opposite to the good direction defined in this draft. And given all probability values should be in the range between $[0,1]$, the values should be all negative (based on the current definition from equation 1) or all positive (based on typical NLPD definition). The current reported value of NLPDs has both negatives and positives at the same time, which implies there might be some mistakes in the calculation. Besides, NLPD is only a way to measure if the model is confident in the true label on the test set, while a good model uncertainty should be large in data points far from the training set and small in data points close to the training set. The authors conducted a simple random 80:20 dataset split, which cannot tell what the model should be confident of. I suggested that authors add the correlation between uncertainty and prediction error as an auxiliary metric; however, I can only find one sentence on page 14 that says the Spearson correlation is 0.3, without any details telling what experiments were done and how this value compared to other uncertainty metrics, such as ensemble variance.

We wish to clarify that the NLPD computes the negative log-likelihood of the test data given the predictive distribution. The log-likelihood here is the sum over log-predictive densities, which is not same as log-probabilities. The probability density function can take values greater than one (unlike probability), thus the log-predictive density can be positive. At the same time, it can also take values less than 1, thus the log-predictive density can be negative as well. Higher the value of log-likelihood (or lower the value of negative log-likelihood), better is the model fit to the observed data. The NLPD penalizes both over- and under-confident predictions and therefore is suitable for assessing the quality of uncertainty estimates. Additional text is now included in the revised manuscript to avoid the confusion (page 14).

As suggested by the reviewer, we have now included more details in the revised manuscript (page 14). The included text is as follows “We also compared the correlation between absolute prediction error and uncertainty estimates provided by the MorganFP-DKL model and an ensemble of GNNs. The mean and variance of the posterior predictive distribution (equations (5) and (6)) obtained from the MorganFP-DKL model is used as the predicted yield and uncertainty score respectively. For ensemble of GNNs, mean and variance of the predictions across different ensembles is considered respectively for the predicted yield and the uncertainty estimate. The spearman rank correlation coefficient (ρ) is then computed between absolute error and uncertainty of predicted yields on the test set. We found that the MorganFP-DKL model has a correlation coefficient comparable to that of GNN ensemble with $\rho=0.30$.”

R3-C5 4. I asked the authors to add Graph DKL BO to Figure 6, but I can't find it.

As suggested by the reviewer, we have now included the plot for BO using GNN-DKL model to Fig. 6 in the revised manuscript.

REVIEWERS' COMMENTS:

Reviewer #3 (Remarks to the Author):

Thanks authors for the detailed response to the raised concerns; now the manuscript looks much better, and I believe it is ready for publication!